# Copy number-dependent DNA methylation of the *Pyricularia oryzae* MAGGY retrotransposon is triggered by DNA damage

Ba Van Vu[1,4], Quyet Nguyen[1,4], Yuki Kondo-Takeoka[1], Toshiki Murata[1], Naoki Kadotani[1], Giang Thi Nguyen[1], Takayuki Arazoe [2], Shuichi Ohsato[3] & Hitoshi Nakayashiki [1✉]

Transposable elements are common targets for transcriptional and post-transcriptional gene silencing in eukaryotic genomes. However, the molecular mechanisms responsible for sensing such repeated sequences in the genome remain largely unknown. Here, we show that machinery of homologous recombination (HR) and RNA silencing play cooperative roles in copy number-dependent de novo DNA methylation of the retrotransposon MAGGY in the fungus *Pyricularia oryzae*. Genetic and physical interaction studies revealed that *RecA* domain-containing proteins, including *P. oryzae* homologs of *Rad51, Rad55,* and *Rad57*, together with an uncharacterized protein, Ddnm1, form complex(es) and mediate either the overall level or the copy number-dependence of de novo MAGGY DNA methylation, likely in conjunction with DNA repair. Interestingly, *P. oryzae* mutants of specific RNA silencing components (*MoDCL1* and *MoAGO2)* were impaired in copy number-dependence of MAGGY methylation. Co-immunoprecipitation of MoAGO2 and HR components suggested a physical interaction between the HR and RNA silencing machinery in the process.

[1] Laboratory of Cell Function and Structure, Graduate School of Agricultural Science, Kobe University, Nada Kobe, Japan. [2] Department of Applied Biological Science, Faculty of Science and Technology, Tokyo University of Science, Noda, Chiba, Japan. [3] Graduate School of Agriculture, Meiji University, Kawasaki, Kanagawa, Japan. [4] These authors contributed equally: Ba Van Vu, Quyet Nguyen. ✉email: hnakaya@kobe-u.ac.jp

n eukaryotic cells, the genetic material is packed into a DNA–protein complex called chromatin. The organization of chromatin into higher-order structures governs diverse nuclear functions, such as transcription, recombination, and DNA repair[1]. Heterochromatin is a repressive chromatin state that is characterized by highly condensed DNA, low transcriptional activity, and a low recombination rate. Typically, chemical modifications of chromatin components, such as histone hypoacetylation, methylation of histone H3 at lysine 9 or 27, and DNA methylation, are associated with heterochromatin formation in a wide range of organisms[2,3].

Repetitive sequences, such as transposable elements (TEs), are common targets for heterochromatin formation, which contributes to the maintenance of genome integrity by suppressing deleterious transpositions and chromosomal recombination between repetitive elements. Moreover, the introduction of transgene arrays, even nontransposable sequences, often induces heterochromatin formation in various organisms[4–7]. Thus, the repetitive nature of a sequence rather than some specific sequence motif appears to be necessary to provoke heterochromatin formation. This notion implies that some surveillance systems for repetitive sequences in the genome may generate a signal for heterochromatin formation. However, the molecular mechanisms responsible for such repeat sensing in the genome are poorly understood.

Another repeat-sensing system in the nucleus is homologous recombination (HR), a fundamental cellular process in which two identical or similar DNA strands exchange genetic material. HR produces new combinations of DNA sequences during meiosis and helps to repair DNA damage, such as double-strand breaks (DSBs) and replication fork collapse[8]. The central players in the search for a homologous sequence during HR are the Rad51 family of proteins, a conserved class of enzymes termed recombinases that include UvsX in T4 bacteriophage and RecA in *Escherichia coli*[9]. Rad51 catalyzes the central reactions in HR, homology search, and DNA strand invasion, by polymerizing on single-stranded DNA (ssDNA) to form a nucleoprotein filament, called the presynaptic filament, which serves as a guide to search for homology in duplex DNA[9,10]. Nevertheless, it remains largely unknown whether there is a molecular link between the two homology-sensing systems.

To address this question, in this study, we used the LTR-retrotransposon MAGGY in the phytopathogenic fungus *Pyricularia oryzae* (*Magnaporthe oryzae*) as an experimental model[11]. When introduced via transformation, the MAGGY element was found to be targeted for DNA methylation even in *P. oryzae* isolates that originally did not carry the element, thus serving as a convenient system for studies on the molecular mechanism underlying de novo DNA methylation[12]. The results show that the *RecA* domain-containing HR components such as *P. oryzae* homologs of *Rad51*, *Rad55*, and *Rad57* participates in copy number-dependent de novo methylation of MAGGY, possibly through a RNA-directed DNA methylation (RdDM)-like pathway. This suggests that HR and heterochromatin formation against retrotransposons could share molecular machinery for repeat search at least in part.

## Results

### A transposition-deficient MAGGY mutant is targeted for DNA methylation in a genomic copy number-dependent manner in *P. oryzae*. 
We have previously shown that DNA of the LTR-retrotransposon, MAGGY, is methylated in the original *P. oryzae* host as well as in MAGGY-free *P. oryzae* isolates after the introduction of MAGGY by transformation[13]. Nonetheless, it is technically difficult to precisely determine the relation between

the copy number in the genome and degree of MAGGY DNA methylation because the element autonomously transposes and increases its copy number independently in each host cell. Thus, we first determined whether MGY-ΔRT, a transposition-deficient MAGGY mutant having a 513-bp deletion in the reverse-transcriptase domain[14], is targeted for de novo DNA methylation in a MAGGY-free *P. oryzae* isolate (Br48). Eleven pairs of PCR primers (Supplementary Table 1) were designed to amplify various MAGGY fragments containing a restriction site for a methylation-sensitive enzyme such as *Hap*II (CCGG) or *Sau*3AI (GATC), and DNA methylation of MGY-ΔRT in a *P. oryzae* transformant was analyzed by the methylation-sensitive restriction enzyme quantitative PCR (qMSRE) method[15]. The results revealed DNA methylation of MGY-ΔRT in every fragment examined with a slight enrichment in the LTRs and 5′ half of the element (Supplementary Fig. 1a–c).

Then, the degree of DNA methylation in the *gag* region and the genomic copy number were determined in 110 *P. oryzae* transformants with MGY-ΔRT. The findings indicated that the degree of DNA methylation significantly correlated with the genomic copy number of the element ($r^2 = 0.545$, $p = 1.56e{-}20$; Fig. 1a). The averages of genomic copy numbers and rates of MAGGY DNA methylation were 6.4% and 5.1%, respectively. In transformants with more than 10 copies of MGY-ΔRT, the rate of DNA methylation was 10.6%, which was comparable to the level of DNA methylation of wild-type MAGGY (average 12.5%, $n = 10$). After that, we used a GFP-expressing plasmid to test whether a foreign gene other than a TE could also trigger copy number-dependent DNA methylation. By contrast, the degrees of DNA methylation of the GFP sequence were much lower than those of the MAGGY sequence regardless of the genomic copy number (Fig. 1a), suggesting that MAGGY specifically induces copy number-dependent DNA methylation in *P. oryzae*.

We further performed whole-genome bisulfite sequencing (Bis-seq) to verify the data of qMSRE. Two *P. oryzae* transformants with ~2 copies (low copy, LC) and 10 copies (high copy, HC) of pMGY-ΔRT were subjected to Bis-seq. Based on the analysis of mapped reads, 0.20% (98/47933) and 2.30% (3128/135720) of C resides in both strands of the original MAGGY sequence were methylated in the LC transformants and the HC transformants, respectively. The rate of 5mC was considerably different depending on sequence context, especially in the HC transformant (Table 1). The average rates of 5mC at CpG were 0.29% and 5.05% in the LC and HC transformants, respectively, which were basically consistent with the rates of MAGGY DNA methylation measured by qMSRE (0.24% in the LC transformant and 9.21% in the HC transformant). Notably, CpA was a more preferred substrate for 5mC over CpC and CpT (Table 1). In higher eukaryotes, symmetric CpNpG contexts were also preferred for 5mC but this is not the case with the MAGGY sequence (Table 1). In the HC transformant, 5mC was detected at 595 sites in both strands of the MAGGY sequence (5638 bp in length). The rate of 5mC at a site was 11.63% in average but it ranged from 4.17 to 34.78% (Supplementary Fig. 1c). The degrees of methylation appeared to be a little higher in 5′ half of the element as shown by qMSRE (Supplementary Fig. 1b, c). It is to be noted that the rate of 5mC was considerably different among reads in the HC transformant. This may be due to a genomic position and/or local genomic structure at the integration sites of pMGY-ΔRT.

In addition, thirteen transformants bearing various copy numbers of MGY-ΔRT were analyzed by conventional bisulfite sequencing (BS). An ~600 bp fragment in the reverse-transcriptase domain was targeted for this analysis. The results indicated that the rates of 5mC measured by qMSRE and BS methods were correlated with each other although the qMSRE

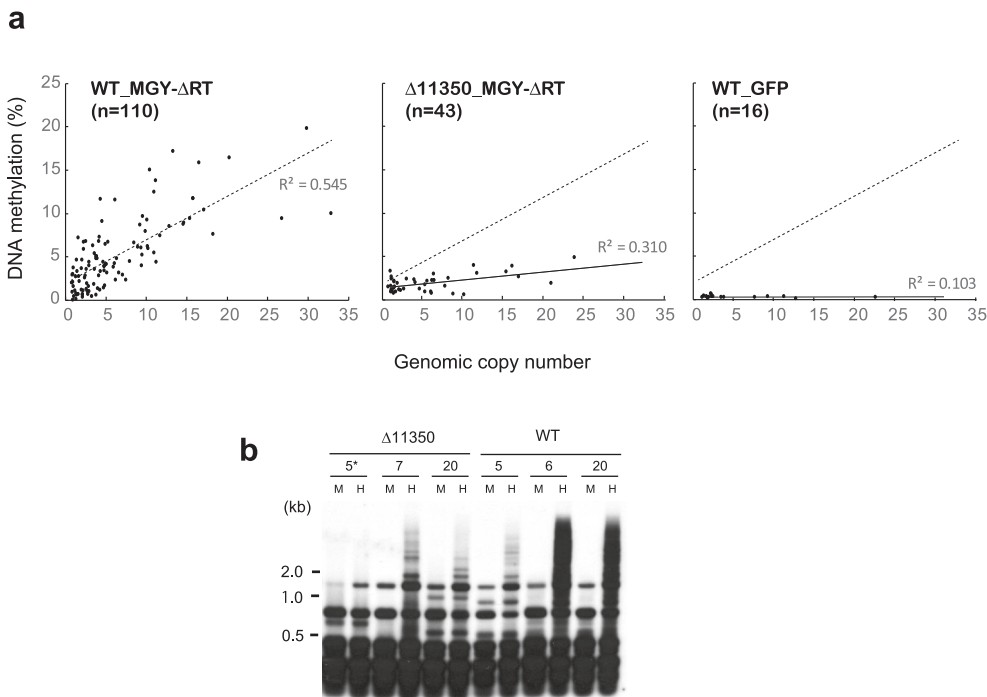

**Fig. 1 Relation between the copy number in the genome and degree of MAGGY DNA methylation in *Pyricularia oryzae*. a** A transposition-deficient MAGGY mutant, MGY-ΔRT was introduced into a MAGGY-free P. oryzae strain, Br48 (WT) and a double deletion mutant of *Rhm51* and *MGG_15577* (Δ11350). The integrated copy numbers and the rates of DNA methylation of MGY-ΔRT were assessed by qPCR and qMSRE methods with a *Hap*II (CCGG) restriction enzyme, respectively. As a control, pEGFP75 carrying the EGFP gene driven by the *Aspergillus nidulans* TrpC promoter was introduced into Br48 (WT_GFP). The dashed line represents the regression line in WT_MGY-ΔRT. **b** Southern blot analysis of DNA methylation of MGY-ΔRT. Three transformants having different integrated copy numbers of MGY-ΔRT were picked up from WT and Δ11350, respectively, and subjected to Southern blot analysis using methylation-insensitive *Msp*I (M) and -sensitive *Hap*II (H) restriction enzymes. *, the integrated copy number of MGY-ΔRT.

**Table 1 Methylation frequency at different tri-nucleotide contexts in MAGGY.**

| sequence | LC transformant[a] | | | HC transformant | | |
|---|---|---|---|---|---|---|
| | 5mC[b] | 5mC + C[c] | rate of 5mC | 5mC | 5mC + C | rate of 5mC |
| CAA | 6 | 2598 | 0.23% | 263 | 7293 | 3.61% |
| CAC | 6 | 1994 | 0.30% | 130 | 5605 | 2.32% |
| CAG | 9 | 3453 | 0.26% | 242 | 8863 | 2.73% |
| CAT | 8 | 1799 | 0.44% | 149 | 5108 | 2.92% |
| CCA | 2 | 3007 | 0.07% | 35 | 8148 | 0.43% |
| CCC | 4 | 3688 | 0.11% | 15 | 10609 | 0.14% |
| CCG | 0 | 4226 | 0.00% | 41 | 12424 | 0.33% |
| CCT | 3 | 2942 | 0.10% | 21 | 8775 | 0.24% |
| CGA | 5 | 3402 | 0.15% | 453 | 10210 | 4.44% |
| CGC | 4 | 3351 | 0.12% | 417 | 10231 | 4.08% |
| CGG | 25 | 5228 | 0.48% | 773 | 13783 | 5.61% |
| CGT | 9 | 3084 | 0.29% | 521 | 8622 | 6.04% |
| CTA | 4 | 1633 | 0.24% | 17 | 4697 | 0.36% |
| CTC | 4 | 2109 | 0.19% | 11 | 5955 | 0.18% |
| CTG | 3 | 2880 | 0.10% | 20 | 8388 | 0.24% |
| CTT | 6 | 2539 | 0.24% | 20 | 7009 | 0.29% |
| total | 98 | 47933 | 0.20% | 3128 | 1E + 05 | 2.30% |

[a]Transformants with ~2 copies (low copy, LC) and 10 copies (high copy, HC) of MGY-ΔRT.
[b]Total number of 5mC within a sequence context in mapped reads (original sequence).
[c]Total number of a sequence context occurred on both strands in mapped reads (original sequence).

rates were generally higher than the corresponding BS rates (Supplementary Fig. 1d). This was likely because the qMSRE rates were measured at a GpC context while the BS rates were average values of 5mC at all sequence contexts. Overall, the results of our bisulfite sequencing-based analyses indicated that the data

assessed by the qMSRE method can be used as an index to evaluate the methylation status of the MAGGY sequence.

**The Rad51 homolog in *P. oryzae* is involved in de novo DNA methylation of MAGGY.** To address the molecular mechanism

responsible for the copy number-dependent MAGGY DNA methylation, we focused on the functional similarities between repeat sensing and HR. If some machinery operates to count a copy number of a sequence on DNA, it could contain a component searching for homologous sequences in the genome. Thus, such repeat sensing machinery may share a component with the HR machinery. Because Rad51 is regarded as a central player in the search for homologous sequences during HR, double mutants of *Rhm51* (*Rad51* homolog in *M. oryzae*, *MGG_15576*)[16] and the neighboring gene (*MGG_15577*) were constructed. We constructed the double mutants because those genes were annotated as a single gene (*MGG_11350.6*) at the start of this study. Using a conventional targeted gene disruption method, three mutants (Δ11350-22, Δ11350-61, and Δ11350-96) were obtained (Supplementary Fig. 2a, b).

In forty-three Δ11350-22 transformants with MGY-ΔRT, the genomic copy number of the element and its degree of DNA methylation were analyzed by qMSRE as described above (Fig. 1a). We also performed Southern blot analyses of representative transformants to verify the qMSRE data (Fig. 1b). The methylation rate of MGY-ΔRT in Δ11350-22 was dramatically reduced compared to that in the wild-type strain (Fig. 1a, b). The average genomic copy number and rates of MAGGY DNA methylation were 5.7% and 1.9%, respectively. Analysis of covariance (ANCOVA) showed that the difference in methylation rates between the wild-type and Δ11350-22 strains was significant ($p = 6.00e{-09}$), indicating positive involvement of *Rhm51* and/or *MGG_15577* in MAGGY methylation in *P. oryzae*. In Δ11350-22, copy number dependence of MAGGY methylation was also decreased but still statistically significant ($r^2 = 0.310$, $p = 7.13e{-5}$; Fig. 1a). A similar reduction in the degree of MAGGY DNA methylation was observed in the other deletion mutants: Δ11350-61 and Δ11350-96 (Supplementary Fig. 2c). We also generated Δ11350-22 transformants with wild-type MAGGY. Consistent with the results on MGY-ΔRT, the average rate of DNA methylation of wild-type MAGGY decreased to a level similar to that of MGY-ΔRT in Δ11350-22 (Supplementary Fig. 3).

To determine which gene, *Rhm51* or *MGG_15577*, is responsible for the decreased level of MAGGY DNA methylation in Δ11350-22, we conducted qMSRE analyses using single-deletion mutants of *MGG_15577* or *Rhm51*. Of note, the results indicated that each of the single disruptants had a phenotype intermediate between the wild-type and the double mutant strains in terms of the level of DNA methylation of MGY-ΔRT (Supplementary Fig. 4c), suggesting that *Rhm51* as well as *MGG_15577* may be synergistically or additively involved in copy number-dependent DNA methylation of MAGGY in *P. oryzae*. Based on these results, we named *MGG_15577* as *Ddnm1* (deficient in de novo DNA methylation 1).

To determine whether *Ddnm1* and *Rhm51* are also involved in the chromatin modification associated with DNA methylation, we analyzed H3K9 and H3K27 methylation in MAGGY sequences by a chromatin immunoprecipitation (ChIP)-qPCR assay. As compared to an actin control, the MAGGY loci were highly enriched in H3K9me3 and H3K27me3 (52.2-fold and 11.7-fold, respectively) in the wild-type strain. The enrichment with H3K9me3 and H3K27me3 was considerably lower in all the deletion mutants when compared to the wild type (Supplementary Fig. 5a). Consistently, transcriptional and transpositional activities of MAGGY were significantly upregulated in the Δddnm1/Δrhm51 mutant (Δ11350-22) (Supplementary Fig. 5b, c).

**Ddnm1 interacts with Rhm51 and plays a role in tolerance to DNA-damaging agents**. A BLAST search revealed that homologs of *Ddnm1* are present in a wide range of ascomycete fungi but apparently not in other classes of organisms. *Rhm51* and *Ddnm1* are syntenic in other ascomycetes but generally separated by the insertion of one or two genes. Ddnm1 shows no significant sequence homology with any functionally characterized protein or motif in the public databases. The growth rates of Δddnm1 mutants were slightly higher relative to the wild-type strain, while the Δrhm51 and the double deletion mutants showed slower vegetative growth (Fig. 2a). Accordingly, we used UV and hydroxyurea (HU) to assess the sensitivity of those deletion mutants to DNA damage. In comparison with the wild-type strain, all the deletion mutants were more sensitive to UV and HU (Fig. 2a). These results indicated that *Rhm51* and *Ddnm1*, with a greater effect of the former, participate in DNA repair, and their contributions to DNA repair could be additive or synergistic.

We then examined whether the Ddnm1 protein could form a complex with the Rhm51 protein in the cell. Two copies of hemagglutinin (HA) or FLAG epitope tags were attached to the N terminus of Ddnm1 and Rhm51, respectively. First, HA-tagged-Ddnm1 and/or FLAG-tagged Rhm51 were introduced into the Δddnm1/Δrhm51 strain. The resulting transformants showed normal growth and tolerance to DNA-damaging agents (Fig. 2a), suggesting that HA-tagged-Ddnm1 and FLAG-tagged Rhm51 were functional. Then, their cell lysates were subjected to immunoprecipitation with an anti-FLAG antibody and subsequently immunoblotted with an anti-HA antibody. Ddnm1 was detectable in the immunoprecipitates in a Rhm51-dependent manner (Fig. 2b, Supplementary Fig. 6). The reciprocal experiment, in which we immunoprecipitated Ddnm1 with the anti-HA antibody, also pulled down Rhm51 (Fig. 2b, Supplementary Fig. 6). The physical interaction between Ddnm1 and Rhm51 was confirmed in the Matchmaker Gold yeast two-hybrid (Y2H) system (Fig. 2c). These results suggested that Ddnm1 formed a complex with Rhm51 in vivo and performed a function in DNA repair and in the copy number-dependent DNA methylation of MAGGY.

**The Rhm51–Ddnm1 complex is not involved in maintenance DNA methylation of MAGGY**. We next tested whether the Rhm51–Ddnm1 complex is involved in de novo and/or maintenance DNA methylation. A *P. oryzae* transformant with 10 integrated copies of MGY-ΔRT was chosen as a parent strain, and then *Rhm51* and *Ddnm1* were knocked out in this strain. The resulting deletion strain showed no significant difference from the parent strain in the degree of MAGGY DNA methylation (Supplementary Fig. 7a).

To further examine the effect of the double Δddnm1/Δrhm51 mutation on genome-wide DNA methylation, we conducted a methylated DNA immunoprecipitation (MeDIP)-seq assay of the wild-type and Δddnm1/Δrhm51 strains carrying 7–8 integrated copies of pMGY-ΔRT (Supplementary Table 2). In the vast majority of genomic regions, the pattern of mapped MeDIP reads was not considerably affected by the Δddnm1/Δrhm51 mutation (Supplementary Fig. 7b). To detect genomic locations showing statistically significantly different levels of DNA methylation between the wild-type and Δddnm1/Δrhm51 strains, the *P. oryzae* reference genome sequence (strain 70-15), which does not have the rRNA cluster and mitochondrial sequences, was divided into 1-kb blocks, and the number of mapped MeDIP reads in each block was determined and subjected to an empirical analysis of DGE (Digital Gene Expression data) (the CLC Genomics Workbench software). Only 15 of the resulting 41,029 genome blocks showed different levels of mapped MeDIP reads between the wild-type and Δddnm1/Δrhm51 strains (Supplementary Table 3). Among the 15 genome blocks, five and two blocks were

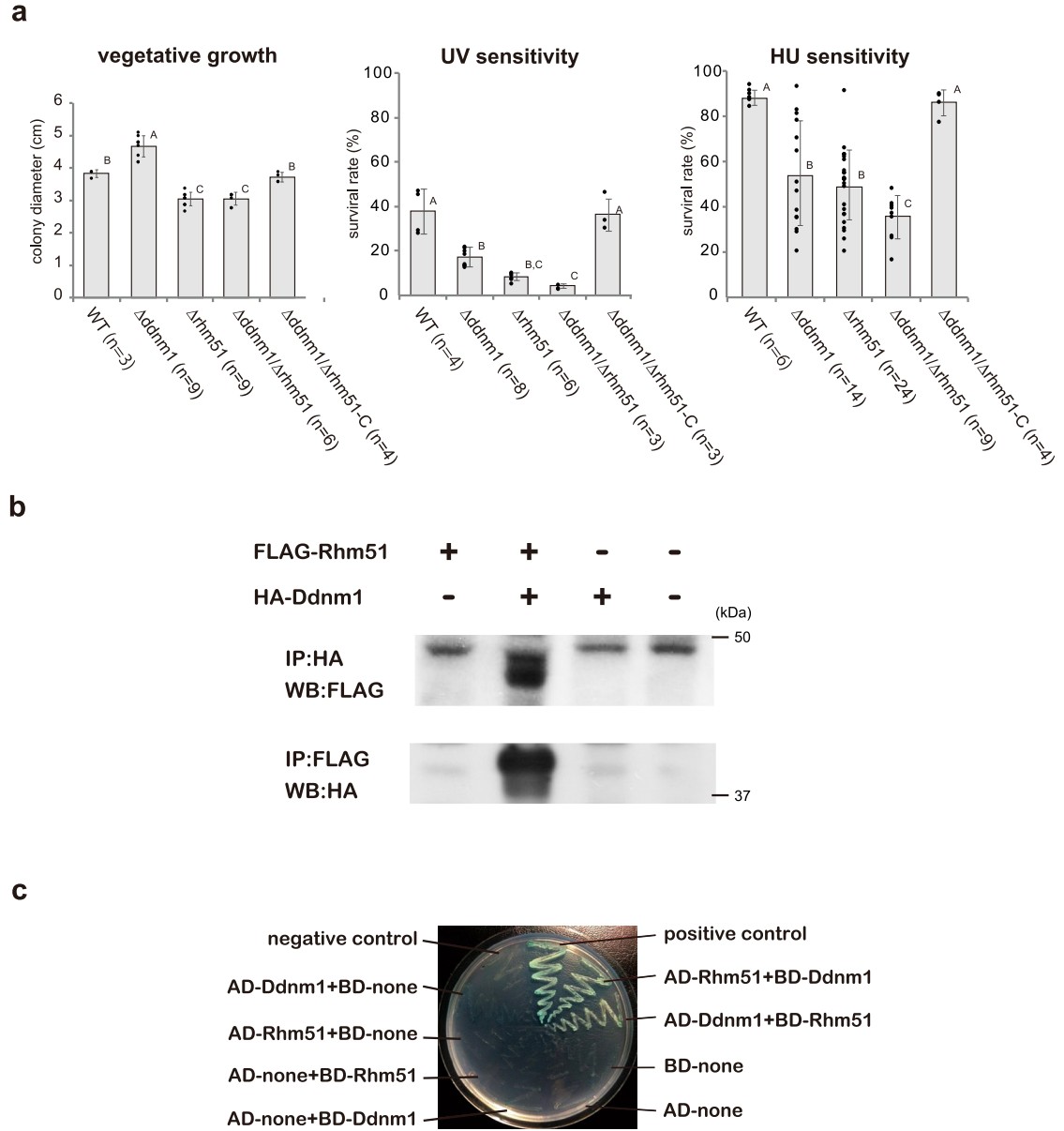

**Fig. 2 Ddnm1 interacts with Rhm51 and plays a role in tolerance to DNA-damaging agents. a** Phenotypic characterization of Δddnm1, Δrhm51 and Δddnm1/Δrhm51 mutants (Δ11350-22), and a gene complementation strain (Δddnm1/Δrhm51-C). For growth assay, colony diameter was measured after 5 days of culture at 26 °C. Sensitivity to DNA-damaging agents (UV and hydroxyurea [HU]) were examined as described in Methods. Each bar represents the average of three or six replicates (±SE). Different letters denote statistically significant differences (Tukey–Kramer HSD test, $P < 0.05$). **b** Co-immunoprecipitation assay of Ddnm1 and Rhm51. Ddnm1 and Rhm1 were tagged with HA and FLAG at the N-terminus, respectively, and introduced into *P. oryzae* cells by the transformation. Cell lysates were immunoprecipitated (IP) with anti-HA or anti-FLAG antibodies, followed by western blotting (WB) with an anti-FLAG or anti-HA antibody. **c** Yeast two-hybrid assay for the interaction of Ddnm1 and Rhm51 was performed using the constructs pGBKT7-53 with pGADT7-T as positive control and pGBKT7-Lam with pGADT7-T as a negative control. The interaction was verified by the growth of mated yeast strains on the SD/−Leu/−Trp/−Ade/−His/+AureobasidinA medium containing X-α-Gal.

contiguous and thus, 10 genomic locations were selected. Among them, three blocks were insertion sites of plasmid vectors such as pMGY-ΔRT in either the wild-type or Δddnm1/Δrhm51 strain, and four blocks contained endogenous MAGGY-like sequences (Supplementary Table 3), indicating that their different levels of MeDIP mapping were not due to the Δddnm1/Δrhm51 mutation itself but due to the introduction of pMGY-ΔRT into the transformants. Thus, the results indicated that only very small fractions of the genome were significantly affected by the Δddnm1/Δrhm51 mutation, indicating that the Rhm51–Ddnm1 complex played a role in de novo but not maintenance DNA methylation.

**de novo DNA methylation of MAGGY is associated with DNA damage**. In gene complementation experiments, we introduced pBS-MoRad51, a plasmid carrying the wild-type *Rhm51* and *Ddnm1* loci, into the Δddnm1/Δrhm51 mutant via either pre-complementation or co-complementation. In the pre-complementation experiments, plasmid pBS-MoRad51 was first introduced into the Δddnm1/Δrhm51 mutant, and then, MGY-ΔRT was integrated into the genome of the resulting transformants, whereas for co-complementation, pBS-MoRad51 and MGY-ΔRT were simultaneously introduced into the Δddnm1/Δrhm51 mutant. In the pre-complementation experiments, the degree of DNA methylation of MAGGY sequences recovered to a

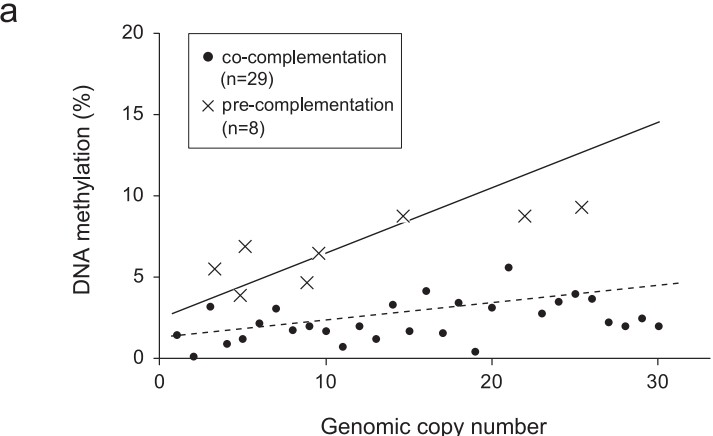

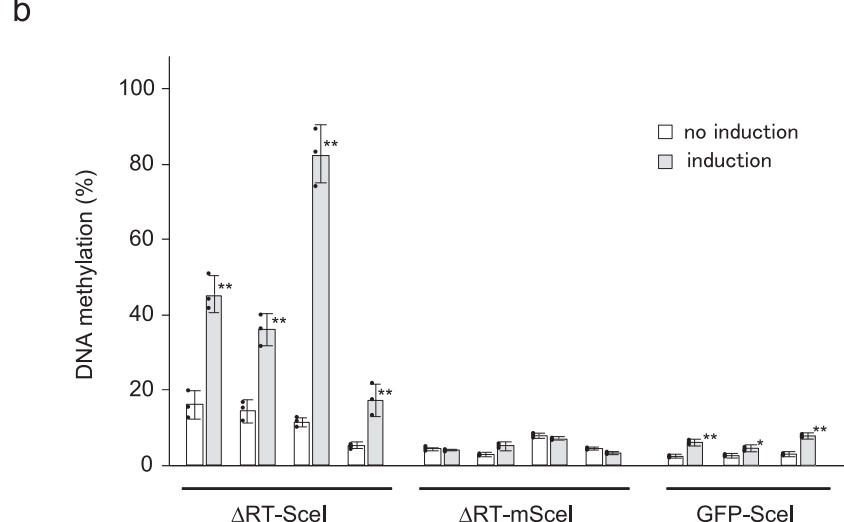

**Fig. 3 de novo DNA methylation of MAGGY is associated with DNA damage. a** Relation between the degree of DNA methylation and genomic copy number of MGY-ΔRT in pre-complementation and co-complementation experiments. In the pre-complementation experiment, pBS-MoRad51 carrying the wild-type *Ddnm1/Rhm51* locus was first introduced into the Δddnm1/Δrhm51 strain. Then, pMGY-ΔRT was further introduced into the resulting gene complemented strain. In the co-complementation experiment, pBS-MoRad51 and pMGY-ΔRT were simultaneously introduced into the Δddnm1/Δrhm51 strain. The solid and dashed lines in the graph represent the regression lines of the wild-type and Δddnm1/Δrhm51(Δ11350) strains shown in Fig. 1a, respectively. **b** The restriction sequence of I-SceI (SceI) or its mutated sequence (mSceI) was introduced in the reverse-transcriptase domain of pMGY-ΔRT or the EGFP coding sequence of pEGFP75. Each of the resulting constructs was introduced into a *P. oryzae* strain possessing pEG2-SceI, a plasmid expressing I-SceI under the control of the cellulose-inducible MoCel7C promoter. The transformants were cultured in CM media containing 1% glucose for 3 days, and transferred to CM media containing 1% cellulose (induction) or 1% glucose (no induction). After culturing for 24 h under inductive or non-inductive conditions, mycelia were harvested and subjected to qMSER analysis. Each bar represents the average of three replicates (±SE). Asterisks denote statistically significant difference from no induction by Student's *t* test (*$P < 0.05$; **$P < 0.01$).

level statistically not different from that in the wild-type strain (ANCOVA $p = 0.44$; Figs. 1a and 3a). In contrast, the degree of MAGGY DNA methylation did not recover in the co-complementation experiments (ANCOVA $p < 0.05$; Fig. 3a). This finding suggests that in order to function as a mediator of MAGGY DNA methylation, the Rhm51–Ddnm1 complex must be pre-assembled before the introduction of pMGY-ΔRT into the host cell by transformation, and consequently, this complex may be necessary for the cell to generate a signal for copy number-dependent DNA methylation during or immediately after the events of integration of the pMGY-ΔRT plasmid into the genome.

On the basis of the gene complementation experiments, we assumed that DSB repair by the Rhm51–Ddnm1 complex at the site of plasmid integration could be directly involved in triggering copy number-dependent de novo DNA methylation of MAGGY. To examine the participation of DSB repair in MAGGY methylation, a DSB was induced into the MAGGY sequence by I-SceI digestion

in vivo. A plasmid expressing I-SceI under the control of the inducible *MoCel7C* promoter[17] and pMGY-ΔRT-Sce, modified pMGY-ΔRT having an I-SceI restriction site in the reverse-transcriptase domain, were introduced into *P. oryzae*. As controls, a *GFP* gene with an I-SceI restriction site and pMGY-ΔRT-mSce having a mutated I-SceI restriction site in pMGY-ΔRT, were also employed in the experiment. pMGY-ΔRT-Sce sequences were methylated at extremely high levels upon I-SceI induction, indicating that MAGGY methylation was triggered by a DSB (Fig. 3b). Such high rates of MAGGY DNA methylation were not detected in transformants with pMGY-ΔRT-mSce even upon induction. DNA methylation of the *GFP* sequence was also induced by I-SceI expression but the degree of *GFP* DNA methylation was significantly lower compared to that of MAGGY in the pMGY-ΔRT-Sce transformants (Fig. 3b). These results suggested that a DSB induces MAGGY-specific DNA methylation in addition to common damage-induced DNA methylation reported in

mammals[18,19]. We also performed this experiment on the Δddnm1/Δrhm51 mutant; however, no stable transformants with the I-SceI-expressing plasmid and pMGY-ΔRT-Sce were obtained, possibly due to lethality because the Δddnm1/Δrhm51 mutant was very sensitive to DNA damage.

**Rad55 and Rad57 homologs also participate in copy number-dependent de novo DNA methylation of MAGGY.** The effects of *Rhm51* and *Ddnm1* on copy number-dependent MAGGY DNA methylation seemed to be additive or synergistic while they form a complex. In addition, the copy number dependence was not completely lost even in the Δddnm1/Δrhm51 mutants (Fig. 1a, c; Supplementary Fig. 2). These observations indicated that additional element(s) other than *Rhm51* and *Ddnm1* could be involved in this process. Therefore, we performed Y2H screening using Ddnm1 and Rhm51 as baits. Two candidates *MoRad55* (MGG_01470) and *MoRad57* (MGG_06985), homologs of yeast *Rad55* and *Rad57* in *P. oryzae*, respectively, were selected for further study because of their sequence similarity to *Rhm51*. Predicted products of *MoRad55* and *MoRad57* consist of 387 and 548 amino acids, respectively, and both possess a recA/Rad51-like domain with Walker A, Walker B, and ATP-binding motifs.

In a single deletion mutant of *MoRad55* (Δmorad55) (Supplementary Fig. 8), copy number dependence of MAGGY methylation was reduced ($r^2 = 0.2053$, $p = 0.005$) compared to that in wild-type ($r^2 = 0.545$, $p = 1.56e{-}20$; Figs. 1a, 4a). This tendency was more remarkable in a double deletion mutant with Ddnm1 ($r^2 = 0.0424$, $p = 0.371$). In contrast, copy number dependency of MAGGY methylation was maintained, albeit decreased, at a significant level in both a single deletion mutant of *MoRad57* (Δmorad57) ($r^2 = 0.3229$, $p = 1.31e{-}4$) and its double mutant with Ddnm1 ($r^2 = 0.5099$, $p = 2.86e{-}6$) (Fig. 4a). Nevertheless, the overall level of MAGGY methylation was much higher in the Δmorad57/Δddnm1 mutant than in the wild-type (Fig. 4a). This tendency was also observed in the single Δmorad57 mutant, especially when the copy number of MAGGY was relatively low. These results suggested that *MoRad57* plays a role in regulating the level of MAGGY DNA methylation while *MoRad55* along with *Rhm51* is more likely involved in repeat-sensing of MAGGY in the genome. To further address the latter point, a double deletion mutant of *Rhm51* and *MoRad55* was constructed. In the resulting Δrhm51/Δmorad55 mutant, copy number dependency of MAGGY methylation was severely compromised and no more statistically significant ($r^2 = 0.0350$, $p = 0.213$) (Fig. 4b), suggesting that *Rhm51* and *MoRad55* function redundantly in the process of repeat-sensing.

To examine the physical interaction among these proteins, we carried out Y2H assay. As shown in Fig. 4c, MoRad55, but not MoRad57, interacted with Ddnm1 while both MoRad55 and MoRad57, albeit at a lesser level in the former, interacted with Rhm51. These results suggested that the Rad51-related proteins could form various complexes to play roles in copy number-dependent MAGGY DNA methylation.

**A specific set of RNA silencing components plays a crucial role in copy number-dependence of MAGGY methylation.** RNA-directed DNA methylation (RdDM), which is mediated by a set of RNA silencing components, is an epigenetic process enabling sequence-specific de novo DNA methylation in eukaryotes. To examine possible involvement of an RdDM-like pathway in copy number-dependent DNA methylation of MAGGY, we employed deletion mutants of two dicer genes (*MoDCL1* and *MoDCL2*) and three Argonaute genes (*MoAGO1, MoAGO2*, and *MoAGO3*) in the analysis. *MoDCL2* and *MoAGO3* were previously shown to be

major players in the RNAi pathway triggered by hairpin RNA, TEs, and mycoviruses in *P. oryzae*[20,21].

The results indicated that copy number dependence of MAGGY methylation was compromised to non-significant levels in Δmodcl1 ($R^2 = 0.014$, $p = 0.454$) and Δmoago2 ($R^2 = 0.002$, $p = 0.752$) but not in the other deletion mutants, suggesting that a specific set of RNA silencing components, other than ones used in the RNAi pathway, plays a crucial role in this pathway (Fig. 5a). However, the coefficient of determination of copy-number dependence of MAGGY methylation was, albeit statistically significant, decreased in Δmodcl2 ($R^2 = 0.188$, $p = 0.0037$) compared to that in the wild-type strain, suggesting that MoDCl2 may also contribute to this pathway to a lesser extent than dose MoDCL1.

In gene complementation experiments, copy number dependence of MAGGY methylation was recovered in both cMoDCL1 and cMoAGO2 strains constructed by pre-complementation. Similar to the complementation experiment with the *Rhm51* and *Ddnm1* genes (Fig. 3a), co-complementation of *MoAGO2* did not restore copy number dependence of MAGGY methylation at a statistically significant level ($R^2 = 0.105$, $p = 0.1138$), suggesting that MoAGO2 also functions as a mediator of copy-number dependent MAGGY methylation during or immediately after the events of integration of the pMGY-ΔRT plasmid into the genome (Supplementary Fig. 9).

Since the components of HR and RNA silencing were both involved in copy number-dependent MAGGY methylation, a physical link between them was examined by co-immunoprecipitation assay. Rhm51, MoRad55, and MoRad57 were tagged with Myc epitope at the C-terminus and introduced into a *P. oryzae* transformant with FLAG-tagged MoAGO2 or the wild-type strain. MoRad55 and Rhm51 were detected in the immunoprecipitates in a MoAGO2-dependent manner (Fig. 5b, Supplementary Fig. 10) but the signal of Rhm51 was weaker than that of MoRad55 (Fig. 5b). These results suggested that the components of HR and RNA silencing interacted with each other to induce sequence-specific copy-number-dependent DNA methylation.

## Discussion

In eukaryotic genomes, TEs are often regulated in a way different from that in other parts of the genome. Typically, they are transcriptionally suppressed by heterochromatin formation and post-transcriptionally silenced by RNAi and/or its related pathways. How TEs are specifically targeted for such gene-silencing mechanisms is a fundamental question that is not yet fully answered. Our present study suggests that the repetitiveness and DNA-damaging nature of TEs are both key factors that trigger effective heterochromatin formation in *P. oryzae*.

Sensing of repeated sequences in the genome is crucial for the maintenance of genome integrity in eukaryotic organisms. As described earlier, heterochromatin formation on dispersed TE sequences and HR also involves homology searching for cognate genomic sequences. In filamentous fungi, repeated sequences are inactivated during the sexual phase of the life cycle by processes called repeat-induced point mutation (RIP) and methylation-induced premeiotically (MIP), which were originally identified in *N. crassa*[22] and *Ascobolus immersus*[23] respectively. Both processes are triggered by repetitive sequences and are associated with DNA methylation, and therefore involve a process of repeat sensing in the genome.

Although these repeat-associated phenomena occur on DNA, the extent of direct DNA–DNA interaction that is responsible for triggering them remains unclear. In *Schizosaccharomyces pombe*, specific repeats designated as *dg* and *dh* repeat elements

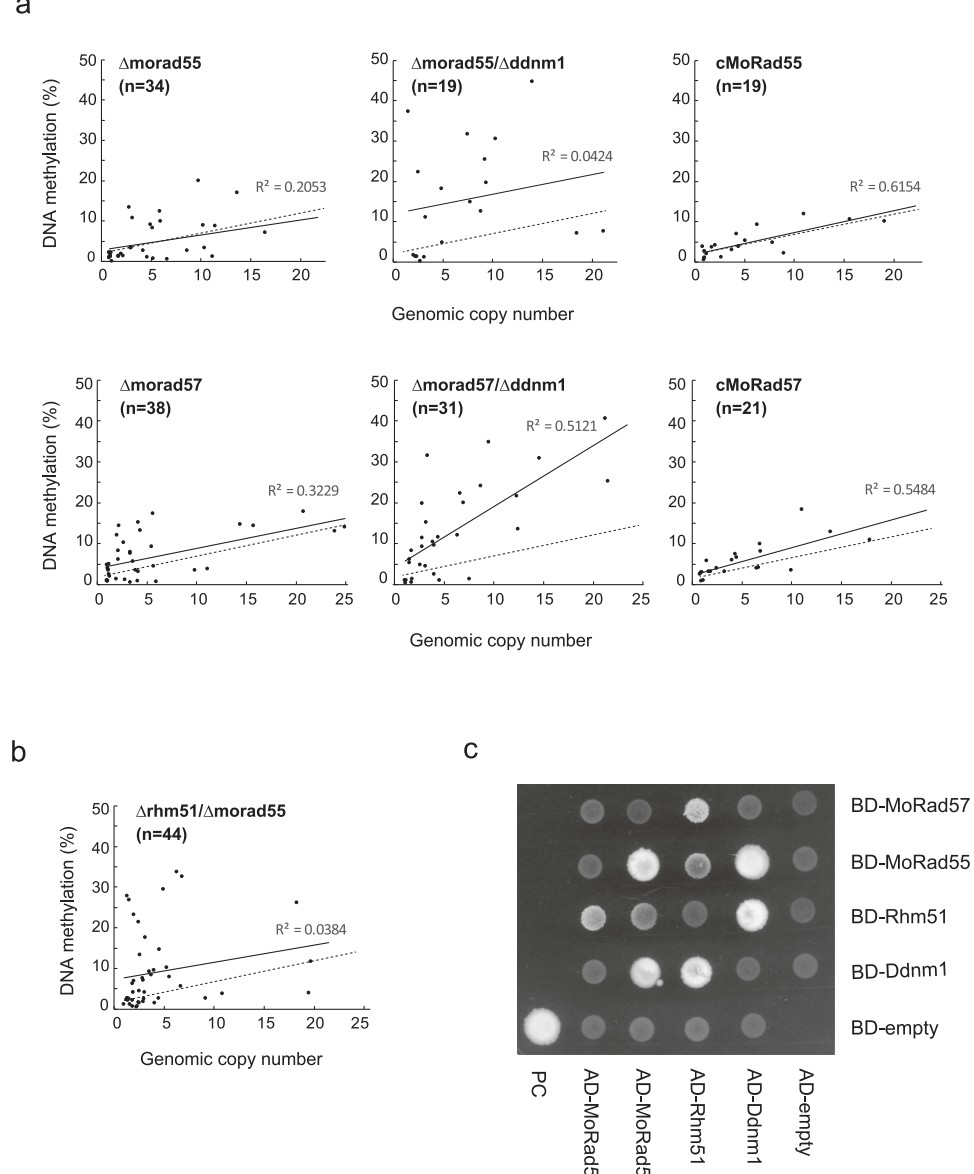

**Fig. 4 Rad55 and Rad57 homologs in P. oryzae participate in copy number-dependent de novo DNA methylation of MAGGY. a** Relationship between the genomic copy number and the rate of DNA methylation of MGY-ΔRT in the Δmorad55 and Δmorad57, and their double mutants with Δddnm1 as well as gene complementation strains (cMoRad55 and cMoRad57). The dashed line represents the regression line of the wild-type strain shown in Fig. 1a. **b** The same assay as in (**a**) was performed with a double deletion mutant of *Rhm51* and *MoRad55*. **c** Yeast two-hybrid assay for interaction among Ddnm1, Rhm51, MoRad55, and MoRad57, was performed using the constructs pGBKT7-53 with pGADT7-T as positive control (PC). Diploid yeast strains expressing each pair of hybrid proteins were grown to equivalent optical density (O.D.600 = 0.8) in 0.5x YPD liquid media, and spotted onto the SD/–Leu/–Trp/-Ade/ –His/+AureobasidinA medium after 20-fold dilution.

invoke the RNA silencing machinery to form constitutive heterochromatin at centromeres, telomeres, and the silent mating-type locus[24,25]. In this model, sequence-specific heterochromatin formation is directed by small RNAs in the RITS complex, and therefore, no machinery for DNA–DNA homology searching is required. In contrast, the RNAi machinery is not required for the maintenance of the heterochromatic state of repetitive sequences in the fungus *N. crassa*[26]. Recently, it was suggested that homology recognition during RIP involves interactions between coaligned double-stranded DNA (dsDNA) molecules but not a RecA (Rad51)-dependent HR machinery[27]. Meanwhile, several components of the HR machinery, such as *Rad18*, *Rad51*, *Rad52*, and *Rad54*, have been implicated in the homology search for small-RNA production from repetitive sequences

in *N. crassa*[28]. Accordingly, different repeat-sensing mechanisms may be responsible for distinct repeat-induced gene-silencing phenomena.

Our present data indicate that copy number-dependent DNA methylation of the retrotransposon MAGGY is mediated by molecular machinery related to *P. oryzae* RecA homologs, *Rhm51*, *MoRad55*, and *MoRad57*, together with the RNA silencing components, *MoDCL1* and *MoAGO2*. These genes affected at least two phenotypic indices: copy number dependence (repeat sensing) and overall level of DNA methylation. *MoDCL1* and *MoAGO2* were mainly involved in copy number dependence, as the average levels of DNA methylation did not differ significantly between their deletion mutants and the wild-type strain, whereas copy number dependence was almost completely abolished in the

a

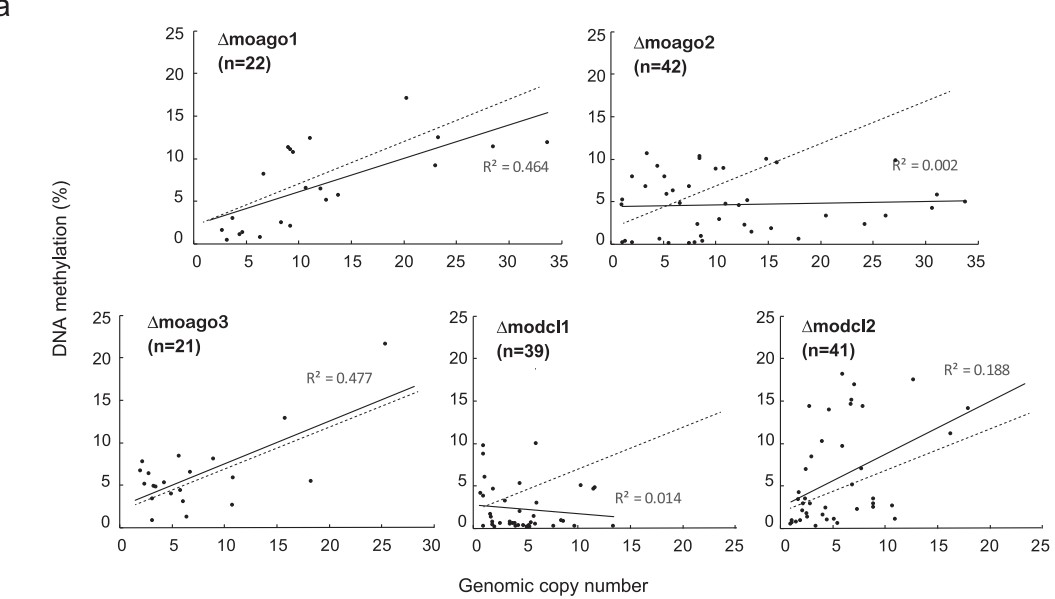

b

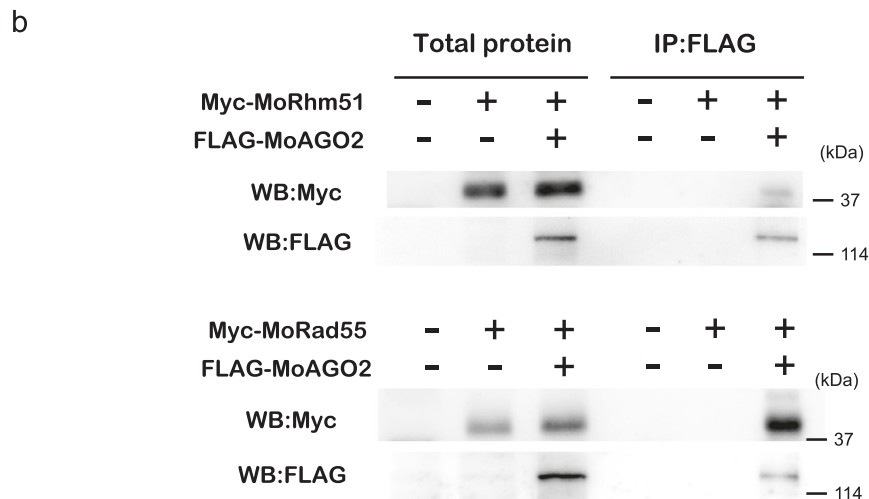

**Fig. 5 A specific set of RNA silencing components plays a crucial role in copy number-dependence of MAGGY methylation in *P. oryzae*. a** Relationship between the genomic copy number and the rate of DNA methylation of MGY-ΔRT in deletion mutants of two Dicer-like genes (Δmodcl1 and Δmodcl2) and three Argonaute genes (Δmoago1, Δmoago2, and Δmoago3). The dashed line represents the regression lines of the wild-type strain shown in Fig. 1a. **b** Co-immunoprecipitation assay of FLAG-tagged MoAGO2 with Myc-tagged MoRhm51 or Myc-tagged MoRad55. Cell lysates (total protein) of *P. oryzae* transformants with these constructs were immunoprecipitated (IP) with anti-FLAG antibodies, followed by western blotting (WB) with an anti-FLAG or anti-Myc antibody.

mutants (Fig. 5a). In *P. oryzae*, another set of RNA silencing components, *MoDCL2* and *MoAGO3*, were responsible for RNAi against TE and mycoviruses[20,21,29]. Of note, RNAi against TEs and mycoviruses was enhanced in Δmoago2[21], suggesting competitive interplay between distinct sRNA-mediated gene regulation pathways in *P. oryzae*. This may be analogous to the RNAi and RdDM pathways in plants, in that each pathway is mediated by a specific set of RNA silencing proteins, and that one is responsible for post-transcriptional gene silencing and the other involves de novo DNA methylation leading to transcriptional silencing[30]. In addition, MoDCL1 produced a slightly longer siRNA than did MoDCL2[31], which is consistent with the use of 21nt and 24nt siRNAs in the RNAi and RdDM pathways respectively, in *Arabidopsis*. However, the evolutionary relationship between RdDM and the MoDCL1/MoAGO2-mediated DNA

methylation pathway needs further investigation since the latter is actually not responsible for DNA methylation itself but rather for repeat sensing. In this context, it is worth noting that MoDCL2 is solely responsible for detectable MAGGY siRNA in vegetative mycelia[29], implying that MoDCL1 dependent siRNA is produced only on some specific events such as DNA damage.

Among the *P. oryzae* RecA homologs, *MoRad55* seemed to contribute to the copy number dependence of MAGGY methylation to a greater extent than did *Rhm51* and *MoRad57*, as only its double deletion mutant with *Ddnm1* was significantly impaired in copy number dependence (Fig. 1a, Fig. 4a). This may be consistent with the co-immunoprecipitation assay where MoRad55 was the most strongly co-immunoprecipitated with MoAGO2, which plays a crucial role in repeat sensing. However, *Rhm51* and/or *MoRad57* likely also have redundant roles in

repeat sensing since copy number dependence was maintained at a statistically significant level in the single Δmorad55 mutant but was severely compromised in the double Δmorad55/Δrhm51 mutant (Fig. 4a, b). Nevertheless, our findings indicated a link between HR and RNA silencing in terms of repeat sensing.

*Rhm51* and *MoRad57* possibly affected the overall levels of MAGGY methylation in opposite directions. The double mutants Δddnm1/Δrhm51 showed a drastic reduction in the overall level of MAGGY methylation (Fig. 1a), whereas the level of MAGGY DNA methylation was generally elevated in Δddnm1/Δmorad57 compared to that in the wild-type strain (Fig. 4a). These data suggest that *Rhm51* and *MoRad57* affected the overall levels of MAGGY DNA methylation positively and negatively, respectively.

The biological function of *Ddnm1* is of interest but is currently unclear. Based on comparisons between single deletion mutants and their double mutants with *Ddnm1*, phenotypic characteristics observed in the single mutant were generally enhanced in the double mutants. This may be explained by assuming that Ddnm1 interacts with several redundantly acting proteins to fully activate them. The relationship between *Rhm51* and *MoRad55* could be such a case.

Intriguingly, *Rhm51* and *Ddnm1* were found to be involved in de novo DNA methylation, but not its maintenance. Our co-complementation and I-SceI digestion experiments suggest that the Rhm51–Ddnm1 complex, and possibly other RecA protein complexes, functions as a mediator of copy number-dependent DNA methylation during the repair of DSBs caused by plasmid integration, enzymatic digestion, or possibly autonomous transposition. In addition, the co-complementation of *MoAGO2* also did not restore copy number dependence (Supplementary Fig. 8). Thus, it is tempting to hypothesize that the homology search for HR and copy number-dependent methylation is performed at the same time after DNA damage by the Rhm51-and-Ddnm1-related machinery, and generates signals for subsequent DNA methylation depending on the extent of repetitiveness, in cooperation with a specific set of RNA silencing proteins. Since repetitiveness and DNA damage are intrinsic natures of TEs, this model provides a general mechanism to explain why TEs specifically trigger heterochromatin formation on their dispersed sequences in the genome.

## Methods

**Fungal strain and transformation**. Fungal strains used in this study were listed in Supplementary Table 4. The fungal transformation was performed as described previously[12]. To introduce wild-type MAGGY and MGY-ΔRT into Br48, pMGY70G and pMGY-ΔRT-G, which carry a geneticin resistance gene and a corresponding element, were used. To screen transformants, hygromycin, and geneticin were used at a concentration of 500 μg/ml in CM media (0.3% yeast extract, 0.3% casamino acid, 0.5% sucrose, 1.5% agar), and blasticidin S was added to Czapek media (0.2% NaNO₃, 0.1% K₂HPO₄, 0.05% KCl, 0.05% MgSO₄, 0.001% FeSO₄, 3% sucrose, 1.5% agar) at a concentration of 100 μg/ml.

**Construction of deletion mutants**. Deletion mutants of Rhm1, Ddnm1, MoRad57, and their double deletion mutants were constructed through targeted gene disruption by HR. The gene disruction vector used was either pSP72-HPH[32], which carries a 1.4 kb hygromycin-resistance gene cassette between the *Sal*I and *Bam*HI sites in pSP72 (Promega), or pII99[33], which carries a geneticin-resistance gene cassette. The 5′ and 3′ flanking fragments of the target gene were amplified by PCR using sets of specific primers (Supplementary Data 1). The amplified fragments were inserted at appropriate sites in either pSP72-HPH or pII99 to establish disruption vectors. The disruption vectors were then introduced into the Br48 strain through PEG-mediated transformation. The resulting *P. oryzae* transformants were screening initially by PCR, and then by Southern blots analysis to confirm a HR event in them (Supplementary Figs. 2, 7).

For gene complementation experiments, a 9.2 kb DNA fragment containing the MGG_15576 and MGG_15577 loci was first PCR-amplified by the high fidelity DNA polymerase, KOD-Plus (Toyobo) with a set of primers (Supplementary Data 1) and cloned into the *Not*I site in pBluescript SK(-) (Stratagene) to establish pBS-MoRad51. pBS-MoRad51 was introduced into the ΔMGG_11350-22 strain by co-transformation with pII99 (pre-complementaion) or pMGY-ΔRT-G (co-complementation).

**Southern blot analysis**. Southern blot analyses were performed as described previously with some modifications[29]. Fungal total DNA was extracted with the Plant Genomic DNA Extraction System (Viogene). For DNA methylation analysis, genomic DNA was digested with a set of isoschizomers (*Msp*I and *Hap*II), and hybridized with a digoxigenin (DIG)-labeled MAGGY probe (a 5.2 kb *Xho*I fragment). Genomic Southern analysis of deletion mutants (Δddnm1, Δrhm51, Δddnm1/Δrhm51, Δmorad55, Δmorad57, Δddnm1/Δmorad55, Δddnm1/Δmorad57) was performed with appropriate restriction enzymes shown in the figure legends and DIG-labeled probes made by PCR with pairs of specific primers (Supplementary Data 1). Prehybridization and hybridization were performed in PerfectHyb™ Plus Hybridization Buffer (Sigma) at 68 °C.

**RNA isolation and quantitative RT-PCR (qRT-PCR)**. RNA isolation and cDNA synthesis were performed as described previously with slight modifications[34]. Total RNA was isolated from frozen mycelial powder using Sepasol RNA I Super G (Nacalai Tesque, # 09379-84). One microgram of total RNA was then subjected to cDNA synthesis using the ReverTra Ace qPCR RT Master Mix with gDNA Remover kit (Toyobo). qRT-PCR assay was carried out using FastStart SYBR Green Master (Roche Applied Science) or GeneAce SYBR qPCR Mix α (Nippon Gene) according to the manufacturer's instructions with specific primers for targets and an internal control actin gene (Supplementary Data 1). Fluorescence from DNA-SYBER Green complex was monitored by Thermal Cycler Dice Realtime System (Takara Bio) throughout the PCR reaction. The level of target mRNA, relative to the mean of the reference housekeeping gene was calculated by the comparative Ct method[35].

**Methylation-sensitive restriction enzyme quantitative PCR (qMSRE)**. The methylation status of MAGGY sequences was analyzed by qPCR using genomic DNA previously digested with the methylation-sensitive enzyme *Hap*II as a template. Four pairs of primers were used in the analysis (Supplementary Data 1): pair A for a MAGGY fragment spanning a *Hap*II site; pair B for a fragment without a *Hap*II site; pair C for a non-methylated region with a *Hap*II site in the *P. oryzae* genome as an enzyme digestion control; and pair D for a single copy genomic region without a *Hap*II site to normalize the data. The rate of DNA methylation was estimated by dividing the normalized quantity value obtained with pair A by the value with pair B. To assess the genomic copy number of MAGGY, the primer pairs B and D were used in qPCR with fungal genomic DNA as a template. Genomic copy number of MAGGY was estimated by dividing the quantity value obtained with pair B by the value with pair D. For methylation analysis of the GFP gene, two pairs of primers (Supplementary Data 1) were used to amplify GFP fragments with and without a *Hap*II site, respectively. This assay was performed just after transformants were obtained.

**Genome-wide and conventional bisulfite sequencing assays**. Fungal total DNA extracted as described above was further purified with DNeasy PowerClean Cleanup kit (QIAGEN). Bisulfite treatment of purified genomic DNA was performed using the EZ DNA Methylation-Gold kit (Zymo Research). For conventional bisulfite sequencing assay, the bisulfite-treated fungal genomic DNA was used as a template to amplify an ~600 bp fragment in the reverse-transcriptase domain of MAGGY with KOD -Multi & Epi- polymerase (TOYOBO) and a pair of primers (Supplementary Data 1). The amplified fragment was cloned into pBluescript SK(+), and ten to fifteen clones were sequenced for each transformant. For genome-wide bisulfite sequencing, pair-end libraries were constructed and sequenced to approximately 7.3 M (HC) and 9 M (LC) reads with 2 × 150PE on Illumina NovaSeq 6000.

**Chromatin immunoprecipitation (ChIP) assay**. ChIP assay was performed using the ChIP-IT™ Express kit (Active Motif, Carlsbad, USA) following the manufacturer's protocol. Fungal mycelia were grown in CM liquid media for 4 days at 26 °C on an orbital shaker (120 rpm). A portion of mycelia (50 mg) was harvested and incubated at room temperature for 15 min in 10 ml of phosphate-buffered saline (PBS) containing formaldehyde at a concentration of 1%. Chromatin was sheared by sonication using a Bioruptor apparatus (Cosmo Bio Co., Ltd., Japan) for 3 cycles of 1 min on at high intensity (200 W) and 30 s off, followed by 4 cycles of 1 min on at medium intensity (160 W) and 30 s off. The size of the sheared chromatin was around 200– 1000 bp as determined by agarose gel electrophoresis. Antibodies against tri-methylated H3 Lys9 (Active Motif, #39161) and trimethylated H3 Lys27 (Active Motif, #39156) were obtained from Active Motif. ChIP-enriched genomic DNA fragments were subjected to qPCR analysis using the primers for MAGGY (pair B, see Supplementary Data 1 and above) and the actin gene (Supplementary Data 1). Input DNA was used to normalize the data. To calculate relative enrichment, the values for MAGGY sequences were divided by the values for the actin gene.

**Immunoprecipitation and immunoblot analyses**. FLAG and HA epitope tags were fused in-frame at the N terminus of Rhm51 and Ddnm1, respectively, by PCR (primers in Supplementary Data 1). The pGT expression vector was used to express the tagged proteins in *P. oryzae* under the control of the *A. nidulans* gpdA promoter[31]. A plasmid expressing FLAG-tagged MoAGO2 was constructed previously[21]. Fungal mycelia of the transformants with the tagged protein construct(s) were ground with mortar and pestle in liquid nitrogen and transferred into lysis

buffer [50 mM Tris-HCl (pH 7.5), 150 mM NaCl, 1 mM EDTA, 0.1% Triton X-100, and 1% Protease Inhibitor Cocktail for use with fungal and yeast extracts (Sigma, #P8215)]. After centrifugation at 12,000 x g for 10 min, the supernatant was mixed with Anti-HA-tag mAb-magnetic beads (MBL, #M180-11) or ANTI-FLAG M2 magnetic beads (Sigma, #M8823), and incubated with gentle agitation for 2–4 h at 4 °C. After washing with cold lysis buffer three times, bound proteins were eluted by boiling for 5 min in SDS sample buffer and subjected to SDS-PAGE. The western blotting analysis was performed using primary anti-FLAG and anti-HA monoclonal antibodies (Wako, # 018-22381 and # 014-21881), and goat anti-mouse IgG-AP conjugate (Bio-Rad, #1706520) as a secondary antibody. Alternatively, Horseradish Peroxidase (HRP) antibody conjugates such as anti-Myc-tag mAb-HRP-DirecT (M192-7, MBL) and anti-DDDDK-tag mAb-HRP-DirecT (M185-7, MBL) were also used. Signals were detected by ImmunoStar Zeta (Wako, #297-72403) using X-ray film or by an image analyzer (ChemiDoc Touch, BioRad).

**Yeast two-hybrid assay**. Yeast culture, transformation, and two-hybrid assays were performed according to the manufacturer's instructions for the Matchmaker Gold Yeast Two-Hybrid System (Clontech). The cDNA fragments of Rhm51 and Ddnm1 to be inserted to pGBKT7 and pGADT7 were obtained by PCR with four pairs of primers (Supplementary Data 1). AH109 and Y187 yeast cells were transformed with the pGBKT7- and pGADT7-constructs, respectively. Mating of two haploid yeast cells of different mating types to create a diploid yeast cell was used to assess the interaction between two test proteins. The control plasmids in the kit, pGBKT7-53 + pGADT7-T as a positive control, and pGBKT7-lam + pGADT7-T as a negative control, were also employed in the analysis. The transformed yeast cells were then plated on SD/-Leu/-Trp agar and SD/-Ade/-His/-Leu/-Trp/X-a-Gal/AbA agar plates.

**DNA damage assays**. To measure sensitivity to DNA damage, conidial suspensions ($1 \times 10^3$ to $10^4$ conidia/ml) of wild-type and mutant strains were divided into halves, and one aliquot was subjected to UV irradiation at a dose of 60 J/m² using CL-1000 ultraviolet crosslinker (UVP), or treated with hydroxyurea (HU) at a concentration of 0.2 mg/ml for 90 min. The treated and non-treated conidial suspensions were mixed with CM media and plated on Petri dishes. To compare survival rates between the treated and non-treated conidia, each plate was incubated at 26 °C for 3 days and the number of colonies that appeared on the plates was counted. This assay was performed with at least three biological replicates.

For I-SceI-induced DSB assay, the recognition sequence of I-SceI, 5′-tagggataacagggtaat-3′, and its mutated sequence, 5′-tagggatcccagcgtaac-3′, were introduced into the EcoRV site of pMGY-ΔRT, establishing pMGY-ΔRT-Sce and pMGY-ΔRT-mSce, respectively. The I-SceI recognition sequence was also introduced into the middle of the eGFP gene in pEGFP75 by inverse PCR using a pair of primers (Supplementary Data 1). The I-Sce-I gene attached with a nuclear localization signal was transferred, by PCR-based cloning, from pGEM-I-SceI_NLS[36] to pEXEG2, a fungal expression vector with the cellulose-inducible MoCel7C promoter[17], and used in transformation experiments.

**Methylated DNA immunoprecipitation, and high-throughput sequencing**. Fungal genomic DNA was extracted as described above. After shearing with the Bioruptor sonicator (Diagenode), heat-denatured methylated DNA was immuno-precipitated with an antibody against 5-methylcytosine (5-mC) (Active motif, # 39649) and Protein A Mag Sepharose (GE Healthcare, #28-9440-06) according to the manufacturer's instructions. DNA SMART ChIP-Seq Kit (Clontech) was used to construct a library for high-throughput sequencing on the MiSeq or HiSeq system (Illumina). Sequencing data were analyzed using Genomics workbench software v11.0 (CLCbio). For MeDIP mapping, the genome of the *Magnaporthe oryzae* strain 70-15 (release 8.0, http://www.broadinstitute.org/) was used as a reference sequence.

**Statistics and reproducibility**. Data are presented as means ± SE for at least triplicate experiments. The linear regression analysis and one-way ANOVA were performed using SPSS version 16.0. Correction for multiple comparisons was performed using the Tukey–Kramer method. Two-tailed unpaired *t* tests were performed using Microsoft Excel.

In ordinary experiments such as phenotypic analysis, we determined the sample size as we have been used, thus, empirically. For correlation analyses between DNA methylation and genomic copy number, the sample size was determined so that we can draw a conclusion by statistical analysis. We basically used all *P. oryzae* transformants obtained in the analysis. However, since transformants with low genomic copies of pMGY-ΔRT (MAGGY) were abundant while ones with a high copy number were rare, some transformants that were supposed to contain only a few copies of pMGY-ΔRT at the stage of PCR screening were not employed for further analysis in some cases.

**Reporting summary**. Further information on research design is available in the Nature Research Reporting Summary linked to this article.

## Data availability
The raw sequence data of MeDIP and GWBS were deposited in the DDBJ/ENA/GenBank database under BioProject ID PRJDB10949. Source data for charts and graphs

in the main figures are provided in Supplementary Data 2. All other data relevant to this study are available from the authors upon reasonable request.

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

## Acknowledgements

We are grateful to Takashi Kamakura (Tokyo University of Science) for the gift of pBF101MBD. This work was supported by a Grant-in-Aid for Scientific Research (A) and (B) from the Japan Society for the Promotion of Science (#25252011 and #25292028).

## Author contributions

H.N. designed the project. H.N., B.V.V., and Q.N. wrote the manuscript. B.V.V., Q.N., S.O., and H.N. analyzed the results. B.V.V., Q.N., Y.K., T.M., N.K., G.T.N., T.A., and H.N. performed the experimental work.

## Competing interests
The authors declare no competing interests.
