## [Peer Review File · Communications Biology]

Reviewers' comments:

Reviewer #1 (Remarks to the Author):

This study shows that copy number-dependent de novo DNA methylation of the retrotransposon MAGGY in *Pyricularia oryzae* is affected by the concerted function of RNA silencing factors and homologous recombination (HR) components. The manuscript is well written and interesting. However, this reviewer has at least 2 major concerns that need to be addressed.

(1) A general comment: the methylation analysis throughout this study was performed by methylation-sensitive restriction enzyme PCR (MSRE). MSRE is an easy, fast and convenient choice for DNA methylation analysis, but I am afraid not the most reliable one. I regret that the authors did not perform the whole analysis by bisulfite sequencing (Bis-seq) which provides the most reliable method of DNA methylation analysis at single base resolution (for an excellent example of Bis-seq analysis in *Magnaporthe oryzae* please see Jeon et al 2015 Sci Rep). Importantly, Bis-seq could provide insight in CG, CHG and CHH methylation. It is necessary to discriminate between which cytosine residue in MAGGY is methylated and which is not. Let me remind here that, at least in plants, CG methylation can be maintained in the absence of RNA trigger thus is not the hallmark of de novo RNA-directed DNA methylation (RdDM), while CHH methylation can not be maintained and constantly requires the RNA trigger, thus being the true hallmark of de novo RdDM. It would thus be very interesting to discriminate between CG/CHG/CHH methylation in MAGGY. These being said, the authors should perform bisulfite sequencing analysis, if not at all, at least in a part of their data, so as to corroborate them. To draw conclusions based only on MSRE is, in my opinion, very risky.

(2) In the last part of their study, the authors investigated the involvement of RNA silencing components in MAGGY DNA methylation. Indeed, they found that DCL1 and AGO2 are involved in the process, and most likely, physical interact with homologous recombination components. Yet, would that also mean that MAGGY small RNAs (siRNAs) produced by DCL1 (and seemingly loaded on AGO2) guide the de novo methyltransferase to MAGGY DNA for methylation? The authors fail to explicitly comment on this important aspect. If this is true, however, then small RNA deep sequencing (sRNA-seq) should be performed. A comparison of sRNA-seq coupled to Bis-seq in DCL1/AGO2 mutants versus wild type could reveal the real involvement of the RNA silencing components in MAGGY DNA methylation and answer urgent questions such as: Are MAGGY siRNAs produced in wild type? Of which size and polarity? Are they homogeneously covering the whole MAGGY body or are they confined to hotspot regions? Does the occurrence of MAGGY siRNAs correlate with the induced cytosine methylation? Which MAGGY siRNAs are eliminated in the DCL/AGO mutants? How is cytosine methylation affected in the absence of siRNAs?

Thank you.

Reviewer #2 (Remarks to the Author):

The study by Nakayashiki and colleagues investigates the role of copy number and DNA damage in controlling DNA methylation of a transposable element in the filamentous fungus *P. oryzae*. The study shows that copy number is correlated with DNA methylation levels. In addition, a role for the homologous recombination machinery (RAD51 and functionally related proteins) and the small RNA processing machinery in controlling DNA methylation levels is reported. The author's further demonstrate a role for DNA damage using an inducible double strand break within a TE construct. Overall, the experiments are well designed and the data support the author's conclusions. These

results are relatively novel within the fungal kingdom and they provide new insights into the determinants of DNA methylation within fungal genomes. Several minor concerns are detailed below.

- Throughout the manuscript, DNA methylation is analyzed primarily by pPCR. In figure 1, a Southern blot is shown to confirm the data. In this blot, a strain with 6 copies appears to have similar methylation levels to a strain with 20 copies. How soon was methylation assayed in these strains following transformation of the MGY-dRT constructs? This is not clear from the methods. Is it possible that 5mC accumulates in these constructs over time and this might also contribute to the difference in 5mC levels?
- In figure 1b, there appears to be significant numbers of MGY-dRT elements with little or no methylation, even in the strain with 20 copies (the MspI bands are clearly evident even in the HapII lane, even in the strain with 20 copies). This suggests that some copies of MGY-dRT are methylated while others remain unmethylated. It is possible that the presence of 5mC is influenced by the genomic position of the integrated retroelement or by the structure of the integration events. For example, in *N. crassa* tandem duplications of transforming DNA more efficiently trigger 5mC. By increasing copy number, the likelihood of such events is higher. It may be difficult to test this here, but these possibilities should be discussed.
- Figure 3b – what is the copy number of the MGY-dRT-SceI construct?

Reviewer #3 (Remarks to the Author):

In the manuscript entitled 'Components of homologous recombination and RNA silencing play cooperative roles in copy number-dependent de novo DNA methylation of a retrotransposon', Nakayashiki and colleagues investigate mechanisms of transposon restriction in the fungus *Pyricularia oryzae*. The authors identify a genetic interaction between the homologous recombination machinery and de novo DNA methylation of retrotransposons at the example of the MAGGY. Next, they show a physical and functional association between the HR protein Rhm51 and Ddnm1 that impacts the response to DNA damaging agents. Finally, the authors uncover an interaction between Rad51 and Ago2 that establishes a link between HR and RNA-dependent DNA methylation. Overall, this project addresses an important aspect of transposon control and connects different molecular mechanisms that collectively protect genome integrity. The experiments are well designed and clearly presented. I recommend publication of this work with minor revision.

Specific comments:

Line 112: 'we focused on the functional similarities between repeat sensing and HR in terms': The authors should elaborate on the similarities and differences.

Fig. 2 A.: The lettering (A, B, C) should be better explained in the Figure legend ('Different capital letters indicate significant differences between the means').

Response to referees' letter

Reviewer #1 (Remarks to the Author):

> Methylation-sensitive restriction enzyme PCR (MSRE) is an easy, fast and convenient choice for DNA methylation analysis, but I am afraid not the most reliable one. I regret that the authors did not perform the whole analysis by bisulfite sequencing (Bis-seq) which provides the most reliable method of DNA methylation analysis at single base resolution. Importantly, Bis-seq could provide insight in CG, CHG and CHH methylation. At least in plants, CG methylation can be maintained in the absence of RNA trigger thus is not the hallmark of de novo RNA-directed DNA methylation (RdDM), while CHH methylation can not be maintained and constantly requires the RNA trigger, thus being the true hallmark of de novo RdDM. It would thus be very interesting to discriminate between CG/CHG/CHH methylation in MAGGY.

As suggested by the reviewer, we have performed whole genome bisulfite sequencing (Bis-seq) as well as conventional bisulfite sequencing analysis (Fig S1c, d; Table 1). The results showed that our qMSRE data can be used as an index of the average 5mC level in the MAGGY sequence. With regard to sequence contexts, the rate of 5mC was considerably higher at CpG and CpA sites than that at the other dinucleotide sites in the MAGGY sequence but no preference for CpNpG was detected (Table 1). 5mC could occur at CHH sites in the MAGGY sequence with lower rates than that at CpG sites.

> In the last part of their study, the authors investigated the involvement of RNA silencing components in MAGGY DNA methylation. Indeed, they found that DCL1 and AGO2 are involved in the process, and most likely, physical interact with homologous recombination components. Yet, would that also mean that MAGGY small RNAs (siRNAs) produced by DCL1 (and seemingly loaded on AGO2) guide the de novo methyltransferase to MAGGY DNA for methylation? The authors fail to explicitly comment on this important aspect. If this is true, however, then small RNA deep sequencing (sRNA-seq) should be performed. A comparison of sRNA-seq coupled to Bis-seq in DCL1/AGO2 mutants versus wild type could reveal the real involvement of the RNA silencing components in MAGGY DNA methylation and answer urgent questions such as: Are MAGGY siRNAs produced in wild type? Of which size and polarity? Are they homogeneously covering the whole MAGGY body or are they

confined to hotspot regions? Does the occurrence of MAGGY siRNAs correlate with the induced cytosine methylation? Which MAGGY siRNAs are eliminated in the DCL/AGO mutants? How is cytosine methylation affected in the absence of siRNAs?

We appreciate reviewer's comments and suggestions very much. Indeed, we performed sRNA-seq analysis of wildtype *P. oryzae* mycelia several times. However, a large amount of sRNAs were mapped to all over MAGGY sequence and no specific mapping was found. We reported previously that another DCL (MoDCL2) is almost solely responsible for biogenesis of MAGGY siRNA in vegetative mycelia (Murata et al. 2007). Thus, we hypothesize that MoDCL1 could be involved in temporal sRNA biogenesis in response to some specific events such as DNA damage. We are addressing this point but have not succeeded in detecting such temporally produced sRNAs to date. Currently, we have no idea whether MoDCL1 and MoAgo2 are simply involved in the RdDM pathway or some other processes as well. The interesting questions raised by the reviewer are surely our future research interests, some of which we are now addressing in the lab. We have added some comments on this issue in the text as below.

In this context, it is worth noting that MoDCL2 is solely responsible for detectable MAGGY siRNA in vegetative mycelia (Murata et al. 2007), implying that MoDCL1 dependent siRNA is produced only on some specific event such as DNA damage.

Reviewer #2 (Remarks to the Author):

> Throughout the manuscript, DNA methylation is analyzed primarily by pPCR. In figure 1, a Southern blot is shown to confirm the data. In this blot, a strain with 6 copies appears to have similar methylation levels to a strain with 20 copies. How soon was methylation assayed in these strains following transformation of the MGY-dRT constructs? This is not clear from the methods. Is it possible that 5mC accumulates in these constructs over time and this might also contribute to the difference in 5mC levels?

We usually perform qMSRE assay immediately after transformation. Thus, approximately two weeks after a recipient cell received the MGY-dRT construct in the genome. We have added a comment on this in the methods. We previously reported that

almost constant level of DNA methylation was maintained over time in *P. oryzae* transformants with the wild-type MAGGY (Nakayashiki et al., 2001).

> In figure 1b, there appears to be significant numbers of MGY-dRT elements with little or no methylation, even in the strain with 20 copies (the MspI bands are clearly evident even in the HapII lane, even in the strain with 20 copies). This suggests that some copies of MGY-dRT are methylated while others remain unmethylated. It is possible that the presence of 5mC is influenced by the genomic position of the integrated retroelement or by the structure of the integration events. For example, in *N. crassa* tandem duplications of transforming DNA more efficiently trigger 5mC. By increasing copy number, the likelihood of such events is higher. It may be difficult to test this here, but these possibilities should be discussed.

We appreciate the comments and suggestions. We did observe both methylated and unmethylated MAGGY sequences in the Bis-seq analysis even in a transformants with high copy number of MAGGY. This may be due to a difference in local genomic structure at the integration sites as suggested by the reviewer. We have added comments on this in the text as below.

It is to be noted that the rate of 5mC was considerably different among reads, especially in the HC transformant. This may be due to a genomic position and/or local genomic structure at the integration sites of pMGY-ΔRT.

> Figure 3b – what is the copy number of the MGY-dRT-SceI construct?

As we stated in the text, we were not able to determine the copy number of the MGY-dRT-SceI construct since this construct was quite frequently lost from the genome. qPCR assay often indicated that the copy number of the construct was below 1 or sometimes even below 0.1 in the genome. This may be explained by that leaky expression of SceI induced DSB in the MGY-dRT-SceI sequence and subsequent loss of the construct during DNA repair.

Reviewer #3 (Remarks to the Author):

> Line 112: ‘we focused on the functional similarities between repeat sensing and HR in terms’: The authors should elaborate on the similarities and differences.

I have added the comments in the text as below.

we focused on the functional similarities between repeat sensing and HR. If some machinery operates to count a copy number of a sequence on DNA, it could contain a component searching for homologous sequences in the genome. Thus, such repeat sensing machinery may share a component with the HR machinery.

> Fig. 2 A.: The lettering (A, B, C) should be better explained in the Figure legend (‘Different capital letters indicate significant differences between the means’).

We have revised the sentence as below.

Different letters denote statistically significant differences (Tukey–Kramer HSD test, $P < 0.05$)

REVIEWERS' COMMENTS:

Reviewer #1 (Remarks to the Author):

I would like to thank the authors for performing the additional experiments which have greatly improved the quality of the manuscript. I happily endorse the manuscript for publication.

Reviewer #2 (Remarks to the Author):

The authors have sufficiently addressed my concerns.

Reviewer #3 (Remarks to the Author):

The authors have addressed all comments and I strongly support publication of this manuscript in the revised form.